# Musculoskeletal Ultrasound Shows Muscle Mass Changes during Post-Acute Care Hospitalization in Older Men: A Prospective Cohort Study

**DOI:** 10.3390/ijerph192215150

**Published:** 2022-11-17

**Authors:** Delky Meza-Valderrama, Ester Marco, Elena Muñoz-Redondo, Andrea Morgado-Pérez, Marta Tejero Sánchez, Yulibeth Curbelo Peña, Elisabeth De Jaime, Lizzeth Canchucaja, Frank Meza Concepción, Stany Perkisas, Dolores Sánchez-Rodríguez

**Affiliations:** 1Rehabilitation Research Group, Hospital del Mar Research Institute, Dr. Aiguader, 88, 08003 Barcelona, Catalonia, Spain; 2Physical Medicine and Rehabilitation Department, National Institute of Physical Medicine and Rehabilitation, Vía Centenario, Diagonal a la Universidad Tecnológica de Panamá, Panama City 0819, Panama; 3Physical Medicine and Rehabilitation Department, Caja de Seguro Social, Calle de Circunvalación, Panama City 0844, Panama; 4Physical Medicine and Rehabilitation Department, Parc de Salut Mar (Hospital del Mar, Hospital de l’Esperança), Sant Josep de la Muntanya 12, 08024 Barcelona, Catalonia, Spain; 5School of Medicine, Universitat Pompeu Fabra, Plaça de la Mercè, 10-12, 08002 Barcelona, Catalonia, Spain; 6Geriatric Department, Centre Fòrum-Hospital del Mar, Parc de Salut Mar, Llull, 410, 08029 Barcelona, Catalonia, Spain; 7Complejo Hospitalario Dr. Arnulfo Arias Madrid, Caja de Seguro Social, Ave. Simón Bolívar, Panama City 07096, Panama; 8University Center of Geriatrics, Antwerp University, Universiteitsplein 1, 2610 Antwerp, Belgium; 9First Line and Interdisciplinary Care Medicine, University of Antwerp, Universiteitsplein 1, 2610 Antwerp, Belgium; 10Geriatrics Department, Brugmann University Hospital, Université Libre de Bruxelles, Place A. Van Gehuchten 4, 1020 Brussels, Belgium; 11WHO Collaborating Centre for Public Health Aspects of Musculoskeletal Health and Ageing, Division of Public Health, Epidemiology and Health Economics, University of Liège, Place du 20 Août 7, 4000 Liege, Belgium

**Keywords:** muscle thickness, cross-sectional area, muscle-skeletal ultrasound, comprehensive geriatric assessment, post-acute care, older adults

## Abstract

This study aimed to prospectively assess changes in muscle thickness (MT) and the cross-sectional area (CSA) of the rectus femoris (RF) muscle in a cohort of older adults, using musculoskeletal ultrasound at admission and at a 2-week follow-up during hospitalization in a post-acute care unit. Differences in frailty status and correlations of MT-RF and CSA-RF with current sarcopenia diagnostic criteria were also studied. Forty adults aged 79.5 (SD 9.5) years (57.5% women) participated, including 14 with frailty and 26 with pre-frailty. In the first week follow-up, men had a significant increase in MT (0.9 mm [95%CI 0.3 to 1.4], *p* = 0.003) and CSA (0.4 cm^2^ [95%CI 0.1 to 0.6], *p* = 0.007). During the second week, men continued to have a significant increase in MT (0.7 mm [95%CI 0.0 to 1.4], *p* = 0.036) and CSA (0.6 cm^2^ [95%CI 0.01 to 1.2], *p* = 0.048). Patients with frailty had lower values of MT-RF and CSA-RF at admission and during the hospitalization period. A moderate-to-good correlation of MT-RF and CSA with handgrip strength, fat-free mass and gait speed was observed. Musculoskeletal ultrasound was able to detect MT-RF and CSA-RF changes in older adults admitted to a post-acute care unit.

## 1. Introduction

Sarcopenia and frailty are strong predictors of morbidity, disability and death in older adults [1] Sarcopenia is a muscle and nutritional disease characterized by a decline in muscle strength and mass, rooted in adverse muscle changes that accrue throughout a lifetime [2,3]. Frailty syndrome is a clinical condition characterized by low muscle mass and strength [1], and an excessive vulnerability of the individual to endogenous and exogenous stressors [4,5]. Although both conditions are associated with extended hospitalization, even short periods of bed rest due to acute processes can lead to muscle loss and functional decline in older adults [6,7,8,9]. However, this muscle loss and functional decline can be reversed when early identification and tailored therapeutic approaches are applied in the early stages [1,10,11]. Sarcopenia may coexist with other age-related diseases that share some phenotypically overlapping features in frail older people and should be adequately assessed and treated [12]. Herein lies the importance of including an assessment of muscle mass and function in the comprehensive geriatric assessment as the basis for planning therapeutic interventions of geriatric medicine and for research purposes [13,14].

Computed tomography and magnetic resonance imaging (MRI) are the gold standard for muscle assessment [15,16], and dual X-ray densitometry (DXA) and electrical bioimpedance analysis (BIA) are the methods most commonly used in clinical practice. However, these technologies require certain levels of technical expertise and are not always available in clinical settings. Ultrasound has emerged as a reliable and valid tool to assess muscle quality and quantity in older populations [17,18], showing stronger correlations when compared with MRI [19], and is suitable for bedside use as part of the comprehensive geriatric assessment [20,21]. The Special Interest Group on Sarcopenia of the European Union Geriatric Medicine Society (EuGMS) has launched an evidence-based initiative aimed at promoting muscle mass assessment to detect sarcopenia through ultrasound (SARCUS) in older people [20,21].

Previous studies have shown good reliability in the ultrasound evaluation of muscles in the lower [22,23] and upper [24,25] extremities. Ultrasound measurements of rectus femoris, as a cross-sectional area (CSA) or muscle thickness (MT), are reliable, simple and robust tools to detect low muscle mass [26,27] and accurately predict sarcopenia in older patients [28,29], with high sensitivity and specificity [30]. CSA and MT are linearly related to maximal voluntary contraction strength in patients with chronic respiratory issues and are also associated with a high to very high prognostic risk in diabetic kidney disease [31]. These measurements assess muscle wasting in hemodialysis patients with protein energy wasting [32] and in patients with spinal cord injury [33]. They also predict adverse outcomes in patients discharged from a surgical intensive care unit [34], and can discriminate between sarcopenia and non-sarcopenia states [34,35]. However, evidence about the longitudinal changes in MT and CSA of the rectus femoris, assessed by ultrasound, remains unavailable in older hospitalized patients. These patients are particularly vulnerable to loss of muscle mass and function during acute diseases as a result of protein undernutrition, inflammatory (pro-catabolic) status or lack of physical activity [36,37].

Based on these considerations, we hypothesized that musculoskeletal ultrasound imaging of the rectus femoris is helpful to detect changes in muscle mass in older adults hospitalized in a geriatric post-acute care ward following a hospital stay for acute illness. The main objective of this study was to use musculoskeletal ultrasound to prospectively assess changes in MT and CSA of the rectus femoris muscle in older patients admitted to a post-acute care unit during a 2-week follow-up. The secondary objectives were to compare MT and CSA in patients with frailty and pre-frailty, and to study the correlations of MT and CSA of the rectus femoris muscle with current sarcopenia diagnostic criteria.

## 2. Materials and Methods

### 2.1. Study Design and Setting

The prospective cohort study followed STROBE (Strengthening the Reporting of Observational Studies in Epidemiology) guidelines [38]. The study was conducted in the post-acute care unit of a university hospital in Barcelona (Catalonia, Spain) from October 2019 to March 2021.

### 2.2. Eligibility Criteria

Inclusion criteria: (1) men and women aged 65 years or older; (2) admitted to a post-acute care unit after hospitalization due to acute illness lasting at least 2 weeks; and (3) cognitive status that permitted understanding study procedures and interventions (Mini-Mental State Examination ≥ 21/30) [39]. Patients with pre-existing conditions that may compromise muscle assessment were excluded (i.e., paresis of the lower limbs due to any neurological disorder, systemic connective tissue disorders, hypo- or hyperthyroidism, lower limb edema at thigh level, systemic atrophies primarily affecting the central nervous system, end-stage renal disease and end-of-life diseases in palliative care).

### 2.3. Study Variables

#### 2.3.1. Muscle Thickness and Cross-Sectional Area

The MT and CSA of the rectus femoris of the dominant or uninjured leg were measured using B-mode on the MyLab™ Seven (Esaote, Genoa, Italy) (Figure 1). All measurements were done at maximal muscle bulk and performed by the same experienced investigator, following the SARCUS protocol for evidence-based muscle assessment though ultrasound [20,21]. The measurement protocol is described in Table 1. Anatomic landmarks and measuring points are shown in Figure 2. All measures were conducted at admission to the post-acute care unit (baseline) and every seven days until discharge, up to a 2-week follow-up.

#### 2.3.2. Muscle Strength

Handgrip strength (Kg) was assessed with the Jamar Plus digital dynamometer, following the Southampton protocol [40]. Patients performed a maximum voluntary isometric contraction of finger flexor muscles. The highest value of three reproducible maneuvers (<10% variability between values) was used for analysis. Values < 27 Kg for men and < 16 Kg for woman were considered as decreased [2].

#### 2.3.3. Muscle Mass

Fat-free mass (Kg) was measured by BIA (Bodystat 1500, Bodystat Ltd., Isle of Man, British Isles, UK). The investigator in charge of assessments performed the BIA immediately after the ultrasound assessment, according to current recommendations [20,41] in a comfortable area with no metallic objects; no meals or drinks 3 h before measurements; no exhausting exercise 12 h before measurements; and no alcohol or caffeine consumption 24 h before measurements [42]. Throughout the examinations, all subjects held their arms and legs in the abduction position; in obese subjects, a towel or pillow was placed between the thighs to avoid skin contact [41]. The impedance values were obtained at a frequency of 50 kHz. Values < 80% of the reference values for the European population [43] were considered as as decreased.

**Table 1 ijerph-19-15150-t001:** The protocol of Muscle Thickness and Cross-Sectional area of rectus femoris measure using musculoskeletal ultrasound.

	Description
Muscle to assess	Rectus femoris of quadriceps
Patient position	Patient lying supine, hips and knees in a neutral position
Patient condition	The patient had to maintain the same position for at least 30 min before the assessment, measuring the muscle in a relaxed state and before any functional testing.
Ultrasound and probe characteristics	B-mode ultrasound, 12 MHz, with a 5 cm-linear transducer probe.
Probe position	The probe in neutral/perpendicular to the skin, with a generous amount of transmission gel, maintaining the minimal pressure possible between the transducer and the skin.
Anatomical landmarks	Proximal landmark: greater trochanter. Distal landmark: proximal border of the patella.
Measuring point	The middle point of distance between anatomical landmarks, marking this point with a demographic pencil.
Measurements procedures	With the transducer probe in the measuring point. The measure of Muscle Thickness: - The most extensive distance from superficial aponeurosis to deep aponeurosis (distance in mm). The measure of cross-sectional area: - The circumference of the muscle, manually drawing with the ultrasound cursor (quadrat area in cm^2^).
Final value	The investigator repeated each measurement three times and used the mean value.

#### 2.3.4. Physical Performance

Physical performance was estimated by gait speed assessed in a 4 m walking test, where values ≤ 0.8 m/s were considered as low gait speed. Participants were instructed to stand with both feet touching the starting line and to begin walking with usual aids (canes or walkers) at their usual pace after a verbal command [44]. Gait speed was considered 0 m/s in bedridden patients unable to stand or in those unable to walk safely.

#### 2.3.5. Sarcopenia Risk

The validated Spanish version of the Strength, Assistance in walking, Rise from chair, Climb stairs, and Falls (SARC-F) questionnaire was administered to assess sarcopenia risk at admission in the post-acute care unit. SARC-F scores range from 0–10 (0 = best to 10 = worst) where a score ≥ 4 is indicative of sarcopenia risk [45,46].

#### 2.3.6. Sarcopenia

Sarcopenia was diagnosed at baseline according to the revised European consensus on definition and diagnosis (EWGSOP2) (2): low muscle mass and low muscle strength. These criteria were applied in all participants, independently of the score obtained in the SARC-F questionnaire.

**Figure 1 ijerph-19-15150-f001:**
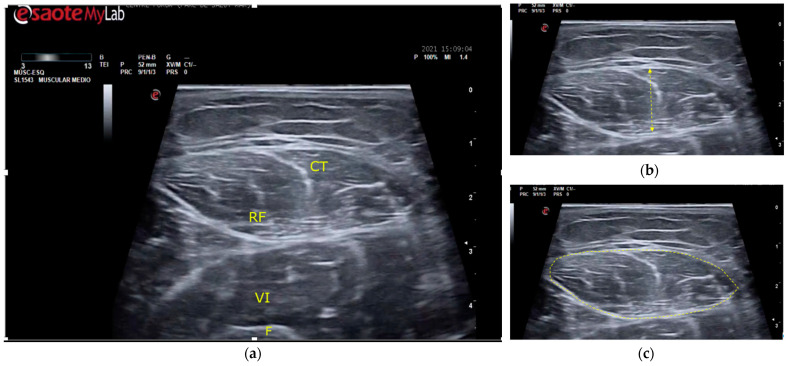
Ultrasound of the rectus femoris muscle for MT and CSA measurement points. (**a**) Guiding structures are shown for proper recognition. RF, rectus femoris muscle; CT, central tendon of RF; VI, vastus intermedius muscle; F, femur. (**b**) MT (muscle thickness) was determined as the distance between the superior and inferior aponeurosis of the RF. (**c**) CSA (cross-sectional area) was measured by delimiting the cross-sectional area of the RF.

**Figure 2 ijerph-19-15150-f002:**
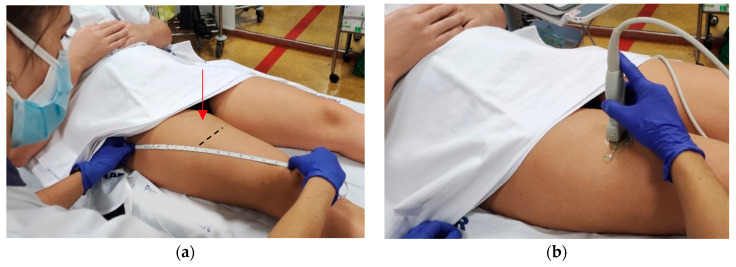
Measuring points of thickness and cross-sectional area of the rectus femoris muscle according to the SARCopenia through UltraSound (SARCUS) protocol [20,21]. (**a**) MT and CSA of RF were measured at the dominant thigh at 50% (red arrow) of the distance between the greater trochanter and the proximal edge of the patella, with the patient’s legs in neutral position. The black line indicates where the images should be captured.; (**b**) A 5 cm width, 12 MHz linear transducer was placed perpendicular to the muscle. Abbreviations: CSA: Cross-sectional area, MT: muscle thickness, RF: Rectus femoris muscle.

#### 2.3.7. Frailty Status

Frailty was assessed by the FRAIL scale at baseline in the post-acute care unit. The FRAIL questionnaire includes 5 components: Fatigue, Resistance, Ambulation, Illness and Loss of weight [47]. Scores range from 0–5 points (0 = best to 5 = worst) and indicate frailty (3–5), pre-frailty (1–2), and robust (0) health status [48].

#### 2.3.8. Nutritional Risk

The Mini-Nutritional Assessment Short Form (MNA-SF) [32] was used to assess baseline malnutrition risk at admission to the post-acute care unit. Scores < 11 indicate risk of malnutrition [49,50].

#### 2.3.9. Laboratory Values

Nutritional values were screened through albumin and prealbumin, and inflammation through C-reactive protein (CRP); vitamin D levels were also evaluated. A blood sample was collected on the day of admission to the unit.

#### 2.3.10. Pharmacological Treatment

Non-steroidal anti-inflammatory drugs and drugs relevant to muscle mass, such as glucocorticoids, allopurinol, statins, insulin, and angiotensin II receptor blockers [51,52,53], were recorded upon admission to post-acute care.

#### 2.3.11. Demographic and Clinical Characteristics

Other collected variables included age (years), sex (male/female), body mass index (Kg/m^2^) and fat mass (Kg). The Charlson Comorbidity Index was used to characterize the population. This international list of comorbidities has a maximum of 36 points, where a score ≥ 2 is an independent risk factor for 1-year mortality risk [54,55]. Falls in the last year were also recorded, as well as previous fragility fractures, defined as a pathological fracture resulting either from minimal trauma or no identifiable trauma at all [56,57]. The diagnosis associated with the functional deterioration that motivated admission to the post-acute care unit was recorded.

### 2.4. Study Procedures

Eligible participants were assessed by the researcher in charge of baseline and weekly assessments. Patients in post-acute care underwent an individualized rehabilitation program according to their functional needs (one hour each day, five days a week during hospitalization, usually a two-week stay). The programs generally focused on improving mobility, starting with the maintenance of joint range, postures, transfers, standing, and walking, followed by progressive muscle strength and endurance exercises.

### 2.5. Sample Size Calculation

Accepting an alpha risk of 0.05 and a beta risk of 0.2 in a two-sided test, 40 subjects were considered as necessary to recognize as statistically significant a change of muscle thickness ≥ 2.6 mm. The standard deviation was assumed to be 5.4. A drop-out rate of 15% was anticipated. The sample size was estimated by using the GRANMO calculator https://www.imim.es/ofertadeserveis/software-public/granmo/, accessed on 1 September 2019.

### 2.6. Statistics

Descriptive analysis was used to determine the clinical and demographic characteristics, using mean and standard deviation for quantitative variables, and absolute values and percentages for categorical variables. The normal data distribution of each variable was estimated with the Kolmogorov-Smirnov test. Chi-square or Fisher tests were used for univariate analysis of categorical variables. Changes in outcomes variables, adjusted by baseline values, were assessed by one-way analysis of variance (ANOVA), and multiple comparison post-test with Dunnett’s test. Intraclass correlation coefficients were calculated to test reliability. Differences in muscle thickness and the muscle cross-sectional area of rectus femoris among pre-frail and frail patients were evaluated with two-way ANOVA. The relationship between the primary and other variables was determined by Pearson correlation coefficient (r), where r ≤ 0.25 indicates absence or minimal relationship; 0.25 to 0.50, low to fair; 0.50 to 0.75, moderate to good; and ≥0.75, strong relationship [58]. The significance level was set at *p* ≤ 0.05. Graphics were processed with GraphPad Prism v.9.3.1 (GraphPad Software LLC, San Diego, CA, USA). Data analysis was performed using the IBM SPSS Statistics v.28 (SPSS Inc., Chicago, IL, USA) software package.

## 3. Results

From a total of 40 participants (aged 79.5 (SD 9.5) years; 57.5% women), 37 completed the baseline and first-week follow-up, 22 completed the 2-week follow-up, and 5 participants completed a 3-week follow-up. Three patients (7.5%) dropped out during the first week of follow-up due to severe deterioration of their health status. Patients had a length of stay in the post-acute care unit of 19.8 days (SD 7.8). Nineteen patients (47.5%) were admitted to the post-acute care unit for rehabilitation after a fracture, four (10.0%) to recover functional deterioration due to infection and seventeen (42.5%) for other causes of functional decline, such as acute illness, exacerbation of preexisting chronic diseases and scheduled surgeries. All participants had frailty (n = 14; 35%; women n = 10) or pre-frailty (n = 26; 65%; women n = 13), and 31 (77.5%) were at risk of sarcopenia. Twenty participants (50%) had low albumin levels, 24 (60%) had low prealbumin, 30 (75%) had low vitamin D levels and 37 (93%) showed high levels of CRP. Baseline demographic and clinical characteristics are displayed in Table 2.

The test-retest reliability of MT and CSA of the rectus femoris using ultrasound was found to be excellent with an intraclass correlation coefficient of 0.970 and 0.968, respectively. Table 3 shows changes in MT and CSA of the rectus femoris, adjusted by their baseline values. In the first week follow-up, men had a significant increase in MT (0.9 mm [95%CI 0.3 to 1.4], *p* = 0.003] and CSA (0.4 cm^2^ [95%CI 0.1 to 0.6], *p* = 0.007). At the end of the second week, men continued to have a significant increase in MT (0.7 mm [95%CI 0.1 to 1.4], *p* = 0.036) and CSA (0.6 mm [95%CI 0.01 to 1.2], *p* = 0.048). Although MT and CSA in women showed an upwards trend throughout the follow-up, these changes were not statistically significant.

Figure 3 shows muscle changes according to frailty status during hospitalization in the post-acute care unit. Frail patients had significantly lower values of both MT and CSA at baseline and during hospitalization: MT mean difference 1.33 mm (95%CI 0.02 to 2.65, *p* = 0.047) and CSA 0.3 cm^2^ (0.29 to 1.47, *p* = 0.004). In the analysis by sex, women in the pre-frailty group had a significantly higher gain in MT (*p* = 0.037; 95%CI: −3.4 to −0.1) and CSA (*p* = 0.034; 95%CI: 0.1 to 1.7) of the rectus femoris muscle, compared to women with frailty.

**Table 3 ijerph-19-15150-t003:** Changes in muscle thickness and cross-sectional area of the rectus femoris muscle assessed through ultrasound during hospitalization in the post-acute care unit according to sex distribution (n = 40).

	Baseline	1st Week Follow-Up		2nd Week Follow-Up	
	Mean (SD)	Mean Difference	*p*	Mean Difference	*p*
		(95%CI)		(95%CI)	
Muscle thickness, mm					
Women	13.2 (2.9)	0.2 (−0.8 to 1.2)	0.932	0.5 (−1.4 to 2.3)	0.847
Men	14.9 (3.4)	0.9 (0.3 to 1.4)	0.003	0.7 (0.1 to 1.4)	0.036
Cross-sectional area, cm^2^					
Women	4.7 (1.5)	−0.1 (−0.5 to 0.3)	0.926	0.2 (−0.5 to 0.9)	0.752
Men	5.3 (1.3)	0.4 (0.1 to 0.6)	0.007	0.6 (0.01 to 1.2)	0.048

Table 4 shows the Pearson correlation coefficients of MT and CSA of the rectus femoris muscle assessed through ultrasound: handgrip strength, fat-free mass and gait speed. A moderate to good correlation of CSA with handgrip strength and fat-free mass was observed at baseline and 1-week follow-up. A low to fair relationship was found between MT, handgrip strength and fat-free mass at baseline and 1-week follow-up (Figure 4). These correlations remained unchanged in the analysis by sex. No relationship was found between MT and gait speed; a low to fair relationship was observed between CSA and gait speed, only at the 1-week follow-up.

## 4. Discussion

This prospective cohort study was aimed to quantify the changes in muscle size estimated with MT and CSA of the rectus femoris muscle, assessed by ultrasound during a 3-week follow-up in patients admitted to a post-acute care unit. The SARCUS protocol for an evidence-based muscle assessment through ultrasound was applied, and the results highlight the need for muscle health assessment during comprehensive geriatric evaluations in patients hospitalized for post-acute care.

Previous studies have reported the adverse effects of hospitalizations on muscle mass and function in older adults [6,11,59]. However, evidence has also shown that early tailored interventions, such as rehabilitation programs, can improve clinical adverse outcomes [59] and shorten hospitalization stays [60]. To the authors’ knowledge, this is the first study to report changes in MT and the CSA of the rectus femoris muscle in an older population hospitalized in a post-acute care unit after acute illness applying the SARCUS protocol.

The MT value, followed by CSA, are considered the simplest and most reproducible ultrasound parameters for muscle mass assessment [61]. Values for rectus femoris MT of 16 mm in women and 20 mm in men have been suggested as cut-off points in the diagnosis of sarcopenia [62]. Although there is no definite consensus on the use of these values, upon admission to post-acute care, our patients presented with a mean MT of 13.2 (SD 2.9) mm in women and 14.9 (SD 3.4) mm in men. By the proposed definition, all would qualify for a sarcopenia diagnosis. Even though the loss of muscle mass during the acute disease could not be quantified in our participants, a significant increase in MT and CSA of the rectus femoris was observed in men after the first week in the post-acute unit, with further changes after the second week. Women showed an upwards trend in MT and CSA throughout the follow-up, but these changes did not reach statistical significance.

In our study, there no were baseline differences in age, comorbidity or frailty status between men and women. A study comparing changes in muscle strength and MT in response to exercise in young adults suggests that MT response in women may have a different time course, with a delay in muscle strength recovery compared to men [63]. However, no clear explanation has been established for these sex-related differences and other factors also may have contributed to the results, such as sex hormones, inflammatory response, sedentary lifestyle and dietary habits, among others [64].

To the authors’ knowledge, this is the first study using musculoskeletal ultrasound as a bedside tool to measure muscle mass as part of the comprehensive geriatric assessment in post-acute care. Musculoskeletal ultrasound was able to detect changes in the 3-week follow-up, and even to differentiate the evolution of muscle changes in frail and pre-frail inpatients. These findings are of important clinical relevance for patients at risk of frailty, which have proven to benefit from interventions that help to improve their muscle mass [65]. Therefore, our study highlights the need for adequate identification of frailty and sarcopenia in this population and supports the importance of including muscle mass monitoring within the comprehensive geriatric assessment in hospitalized older adults [66] to target interventions (e.g., exercise) that enable attenuating or preventing muscle wasting [67].

Several study limitations should be acknowledged. First, the relatively limited sample size, especially for men at the 3-week follow-up, hindered a deeper analysis of MT and CSA changes in the rectus femoris. Second, the limited information about the number of bed rest days in the acute care unit prior to admission in the post-acute care unit precluded analysis of functional impairment before referral to the post-acute unit. Given that MT and the CSA of rectus femoris muscle parameters are highly sensitive to bed rest, those patients who had more days of bed rest during acute hospitalization may have presented with higher initial values than the rest of the patients, affecting the evolution of the MT and CSA changes. Finally, a selection bias must be acknowledged, given the absence of “robust” patients in the sample. These limitations should be addressed in further prospective multicentric studies with larger sample sizes. The study highlights the need to re-think diagnostic strategies in sarcopenia and implement updated evidence-based diagnostic tools as part of a comprehensive geriatric assessment in clinical practice, in an effort to improve the quality of care and remain at the forefront of best practices.

## 5. Conclusions

Musculoskeletal ultrasound was able to detect MT and CSA improvement in men during a 2-week follow-up in a post-acute care unit. Frail patients show lower values of these parameters during the hospitalization when compared to pre-frail patients. MT and CSA measured by ultrasound were correlated with handgrip strength and fat-free mass in this sample at baseline and at one-week follow-up.

## Figures and Tables

**Figure 3 ijerph-19-15150-f003:**
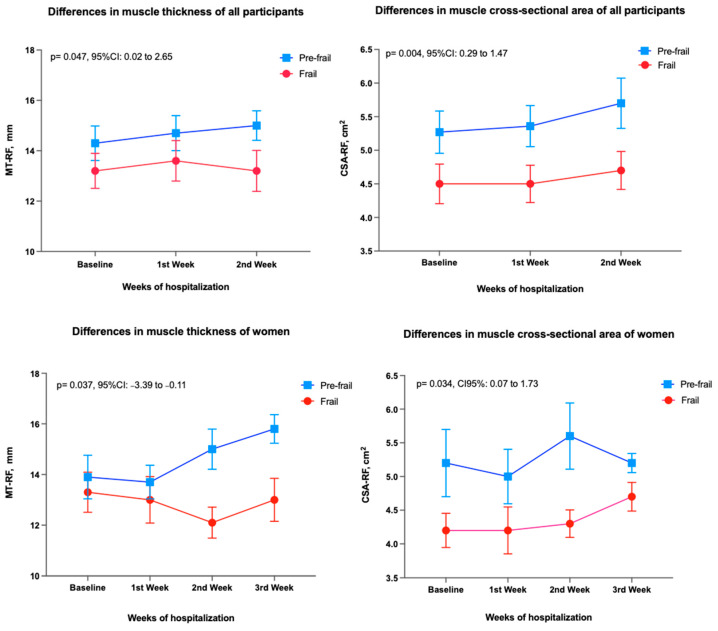
Differences in muscle thickness and muscle cross-sectional area of rectus femoris among groups (Pre-frailty vs. Frailty) evaluated with two-way analysis of variance. The mean values and the standard error of the mean are shown. Abbreviations: CSA: Cross-sectional area, MT: muscle thickness, RF: Rectus femoris muscle, CI: Confidence Interval.

**Figure 4 ijerph-19-15150-f004:**
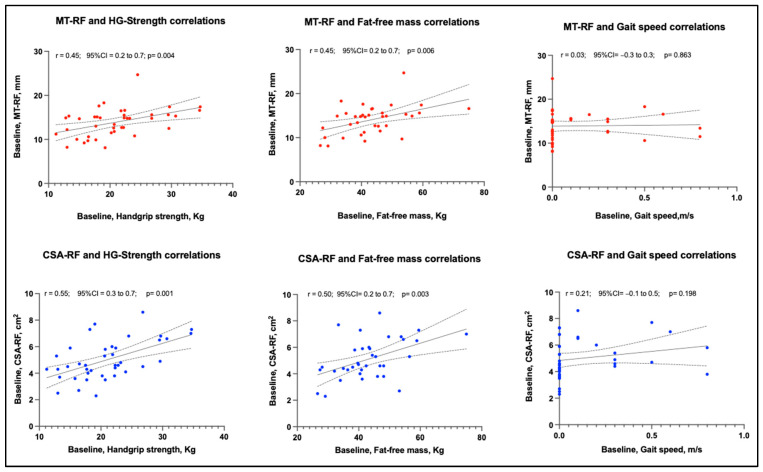
The relationship of baseline muscle thickness and cross-sectional area of rectus femoris with muscle strength (handgrip), muscle mass (fat-free mass) and physical performance (gait speed), assessed during hospitalization in a post-acute care unit. Abbreviations: CSA: Cross-sectional area, HG: Handgrip; MT: Muscle thickness, RF: Rectus femoris muscle. r: the Pearson coefficient, CI: Confidence Interval.

**Table 2 ijerph-19-15150-t002:** Baseline demographic and clinical characteristics of study participants at admission to the post-acute care unit.

	Total Sample (n = 40)
Demographics:	
Age, years (SD)	79.5 (9.5)
Sex, female, n (%)	23 (57.5)
Ultrasound assessment:	
Muscle thickness, mm (SD)	
Women	13.2 (2.9)
Men	14.9 (3.4)
Cross-sectional area, mm^2^ (SD)	
Women	4.8 (1.5)
Men	5.3 (1.4)
Body composition:	
Body mass index, Kg/m^2^ (SD)	29.1 (6.0)
Fat-free mass, Kg (SD)	
Women	38.3 (6.7)
Men	51.1 (10.0)
Fat mass, Kg (SD)	
Women	34.6 (9.6)
Men	26.6 (8.3)
Muscle strength,	
Handgrip, Kg (SD):	
Women	18.1 (3.8)
Men	24.9 (6.1)
Physical performance:	
4 m gait speed test, m/s (SD)	0.12 (0.2)
Sarcopenia assessment:	
SARC-F/10 (SD)	6 (3)
Sarcopenia (EWGSOP2), n (%)	7 (17.5)
FRAIL scale, /5	
Frailty (3–5), n (%)	14 (35)
Pre-frailty (1–2), n (%)	26 (65)
Robust (0), n (%)	0
Malnutrition risk	
MNA-SF, /14 (SD)	10 (2)
Comorbidity	
Charlson Index, /36 (SD)	3 (4)
Laboratory test	
Vitamin D, ngmL (SD)	18.2 (10.1)
Albumin, gdL (SD)	3.4 (0.3)
Prealbumin, mgdL (SD)	18.3 (6.6)
C Reactive Protein, mgdL (SD)	5.1 (4.7)
Falls in the last year, n (%)	
0 falls	12 (30)
1–3 falls	22 (55)
4 or more falls	6 (15)
Fragility fractures, n (%)	19 (47.5%)
Pharmacologic treatment, n (%)	
Glucocorticoids	23 (57.5)
Allopurinol	6 (15)
Statin	14 (35)
Insulin	10 (25)
ARBs	9 (22.5)
NSAIDs	38 (95)

Body mass index (Kg/m^2^; in ≥70-year-olds, reduced if <22 Kg/m^2^); muscle strength assessed by handgrip dynamometer (Kg; low if <27 Kg in men and <16 Kg in women), muscle mass estimated with fat-free mass assessed by bioimpedance analysis (Kg, reduced if <80% of the European reference values); and physical performance assessed by the 4 m gait speed test (m/s, low if <0.8 m/s) according to the revised consensus on definition and diagnosis of sarcopenia (EWGSOP2). Abbreviations: ARBs: Angiotensin II receptor blockers; EWGSOP: European Working Group of Sarcopenia in Older People; FRAIL: Fatigue, Resistance, Ambulation, Illness and Loss of weight; scores indicate frailty (3–5), pre-frailty (1–2) and robust (0); MNA-SF: Mini-Nutritional assessment Short-form, where a score 8–11 indicates being “at risk of malnutrition”; NSAIDs Non-steroidal anti-inflammatory drugs; SARC-F: Strength, Assistance with walking, Rising from a chair, Climbing stairs and Falls questionnaire, where scores range from 0 to 10 and a score ≥ 4 is indicative of sarcopenia; SD: Standard deviation.

**Table 4 ijerph-19-15150-t004:** Pearson correlation coefficients of muscle thickness and cross-sectional area of the rectus femoris assessed by ultrasound with current sarcopenia diagnostic criteria during hospitalization in the post-acute care unit (n = 40).

	Muscle Thickness	Cross-Sectional Area
	r (*p*)	r (*p*)
Baseline		
Handgrip	0.45 (0.004)	0.56 (0.001)
Fat-free mass	0.45 (0.006)	0.50 (0.003)
Gait speed	0.03 (0.863)	0.21 (0.198)
1st week		
Handgrip	0.42 (0.011)	0.58 (0.001)
Fat-free mass	0.42 (0.015)	0.57 (0.001)
Gait speed	0.03 (0.974)	0.38 (0.027)
2nd week		
Handgrip	0.38 (0.080)	0.34 (0.124)
Fat-free mass	0.31 (0.192)	0.44 (0.060)
Gait speed	0.26 (0.238)	0.38 (0.085)

Components of the European Working Group of Sarcopenia in Older People (EWGSOP) criteria: handgrip strength, fat-free mass and gait speed. Pearson correlation coefficient (r), where r ≤ 0.25 indicates absence or little relationship; 0.25 to 0.50, low to fair; 0.50 to 0.75, moderate to good; and ≥0.75, strong relationship [58].

## Data Availability

Not applicable.

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
