# Peer review of "Musculoskeletal Ultrasound Shows Muscle Mass Changes during Post-Acute Care Hospitalization in Older Men: A Prospective Cohort Study"

_ijerph, 2022, doi:10.3390/ijerph192215150_

Round 1

Reviewer 1 Report

I am grateful for the opportunity to review this manuscript titled “Musculoskeletal ultrasound shows muscle mass changes during post-acute care hospitalization in older adults: A prospective cohort study”. The purpose of this study was to use musculoskeletal ultrasound to prospectively assess changes in muscle thickness and cross-sectional area of the rectus femoris muscle in older patients admit- ted in a post-acute care unit during a 3-week follow-up. The data collected in this study may affirm or expand on available literature.

This study is of interest to the IJERPH readers and seems to provide some new findings, applicable to the fields of rehabilitation and training. However, the points mentioned in the “Specific comments” section below should be considered and the manuscript amended accordingly before being considered for publication.

Specific comments

1.     Authors should check the number of pages. From page 7 onwards there is an error.

2.     Authors must respect the format proposed by the journal. In the discussion section, there is a different spacing between paragraphs than in the rest of the document. They also use a different indentation, ...

3.     In the conclusions section, the text is not justified.

Abstract

4.     The authors could briefly describe the measurement protocols.

5.     It would be appropriate for authors to introduce statistical values in the abstract (i.e., p-value, effect size, ...).

Introduction

6.     The introduction is well written, but the authors need to highlight in the introduction the contribution of their work to the area. I recommend expanding the introduction section.

7.     The study hypothesis is missing.

Materials and Methods

8.     Please report your test-retest reliability for all tests used.

9.     Please expand the section "2.3.3. Muscle mass". How were the measurements taken? What condition were the subjects in when muscle mass was assessed (i.e., fasting, ...)?

10.  Please expand the section "2.3.4. Physical performance".

11.  Changes during follow-up should be assessed with a repeated measures test. Not with a Student's t-test. For me, this is a red flag.

Reviewer 2 Report

The current manuscript aimed to assess muscle thickness and CSA of the rectus femoris by ultrasound in a hospital setting among older patients of both sexes over a 3 week period. The findings of the study offered evidence that the SARCUS method employed by the authors could provide measures of muscle parameters during this period of follow up and thereby could represent a cost effective method for assessing muscle status during rehabilitation in a hospital. This is useful information due to the well established observation that sarcopenia is a remarkable risk for elderly hospitalized patients that predisposes such patients to loss of function and independence. The authors also correlated some other measures such as handgrip strength and gait performance with RF muscle parameters. Taken together, these data indicate that ultrasound measures could be useful in tracking patient outcomes and perhaps customizing rehabilitation protocols to preserve muscle function and combat sarcopenia while protecting function and independence in daily living.

The authors should provide more clarity in describing whether or not patients were engaged in therapeutic modalities during the hospitalization period aimed at preserving or increasing RF muscle mass. This reviewer only found in the text that the participants were admitted to a post-acute treatment care unit for the duration of the study. The authors report that there were increases in RF MT among pre-frail subjects and it is unclear whether and what kind of therapy the patients received or if the muscle changes were simply a result of recovery from conditions for which they had been admitted to the post-acute care facility. 

The main finding from the text as written seems to be that SARCUS can assess changes in the RF muscle during a 3 week period of follow up in a hospital setting. 

Author Response

AUTHORS RESPONSE TO REVIEWERS

Thank you very much for your valuable comments and feedback regarding our research paper. We attach a point-by-point response to those comments and the revised manuscript with all changes highlighted.

REVIEWER COMMENT: The authors should provide more clarity in describing whether or not patients were engaged in therapeutic modalities during the hospitalization period aimed at preserving or increasing RF muscle mass. This reviewer only found in the text that the participants were admitted to a post-acute treatment care unit for the duration of the study. The authors report that there were increases in RF MT among pre-frail subjects and it is unclear whether and what kind of therapy the patients received or if the muscle changes were simply a result of recovery from conditions for which they had been admitted to the post-acute care facility. 

AUTHORS’ RESPONSE: Thank you very much for this observation. The participants in this longitudinal study did not participate in any therapeutic modality specifically aimed to preserve or increase muscle mass. Each patient received an individualized rehabilitation program according to their baseline functional condition. We have added this information in the text as follows:

“Study procedures; Eligible participants were assessed by a researcher who was in charge of baseline and weekly assessments. Patients in post-acute care underwent an individualized rehabilitation program according to their functional needs (one hour each day, five days a week during hospitalization, usually two weeks). The programs generally focused on improving mobility, starting with the maintenance of joint range, postures, transfers, standing, and walking, followed by progressive muscle strength and endurance exercises.” (Materials and Methods, lines 213-218).

REVIEWER COMMENT: The main finding from the text as written seems to be that SARCUS can assess changes in the RF muscle during a 3 week period of follow up in a hospital setting. 

AUTHORS’ RESPONSE: Yes, our findings suggest that musculoskeletal ultrasound is a useful tool for detecting changes in muscle mass during a hospitalization period of 2 weeks. In the first version of the manuscript, we described changes in a 3-week follow-up. In the revised manuscript, we have not included data from the third week because most of the patients had already been discharged home.

Reviewer 3 Report

Dear Authors,

thank you for giving me the opportunity to revise your manuscript. The paper aimed to investigate the changes in muscle mass in older adults in post-acute care unit. The topic is very interesting in field and has not only a quite good scientific soundness, but also in clinical practice. Nevertheless, there are some critical issue to be addressed:

Introduction: 

The introduction is well written, but I suggest to improve it. Sarcopenia is one the most important topic in eldery and can be cause of different associated diseases. Please, read "de Sire A, Ferrillo M, Lippi L, Agostini F, de Sire R, Ferrara PE, Raguso G, Riso S, Roccuzzo A, Ronconi G, Invernizzi M, Migliario M. Sarcopenic Dysphagia, Malnutrition, and Oral Frailty in Elderly: A Comprehensive Review. Nutrients. 2022 Feb 25;14(5):982. doi: 10.3390/nu14050982." Moreover, the US are a very easy and feasible tool to evaluate the muscle mass in clinical practice and can predict with a good reliability the muscle mass waste and bone. In this scenario, US of other muscle site can be useful in sarcopenia diagnosis. Please, read "Leigheb M, de Sire A, Colangelo M, Zagaria D, Grassi FA, Rena O, Conte P, Neri P, Carriero A, Sacchetti GM, Penna F, Caretti G, Ferraro E. Sarcopenia Diagnosis: Reliability of the Ultrasound Assessment of the Tibialis Anterior Muscle as an Alternative Evaluation Tool. Diagnostics (Basel). 2021 Nov 21;11(11):2158. doi: 10.3390/diagnostics11112158."

Methods: Please, specify more in detail the inclusion criteria for the patients and the follow-up in this section. Moreover, it could be interesting to know the rehabilitation program that patient underwent in post-acute care. 

Moreover, a punctual demographics characteristics, such as lab test (total protein, vitamin d status etc.)  and pharmacological treatments is mandatory to better define the population. Moreover, have your population experience of falls or anamnestic fragility fracture? please, specify.

The statistical analysis can be improved with analysis more specific to corroborate the association in outcome, such as a multivariate analysis corrected with age, comorbidity and sex. It would be desirable also correction with lab test and pharmacological treatment, if available.

Results: Reading the table 3, It seems that HG, free fat mass and gait speed are inversely associated to cross sectional area  at 3 weeks. Please, specify this point in discussion

Discussion: The discussion is lack of information. I suggest to improve it with a robust literature to support your results and, if applicable, compare them with other similar study (e.g US in sarcopenia).

Many compliments for the papers

Best wishes for your future research

Best regards

Author Response

AUTHORS RESPONSE TO REVIEWERS

Thank you for your very careful review of our paper, and for the thoughtful comments, corrections and suggestions. We attach a point-by-point response to those comments and the revised manuscript with all changes highlighted.

Introduction: 

REVIEWER COMMENT: The introduction is well written, but I suggest to improve it. Sarcopenia is one the most important topic in eldery and can be cause of different associated diseases. Please, read "de Sire A, Ferrillo M, Lippi L, Agostini F, de Sire R, Ferrara PE, Raguso G, Riso S, Roccuzzo A, Ronconi G, Invernizzi M, Migliario M. Sarcopenic Dysphagia, Malnutrition, and Oral Frailty in Elderly: A Comprehensive Review. Nutrients. 2022 Feb 25;14(5):982. doi: 10.3390/nu14050982." 

AUTHORS’ RESPONSE: Thank you for your insight to improve the quality of our manuscript. We have read with attention the suggested review and have incorporated the reference (1) to reinforce the importance of assessment for other sarcopenia-related diseases. We have included the following information:

”(…) Sarcopenia may coexist with other age-related diseases that share some phenotypically overlapping features in frail older people and should be adequately assessed and treated (1). (…)”. (Introduction, lines 59-61).

REVIEWER COMMENT: Moreover, the US are a very easy and feasible tool to evaluate the muscle mass in clinical practice and can predict with a good reliability the muscle mass waste and bone. In this scenario, US of other muscle site can be useful in sarcopenia diagnosis. Please, read "Leigheb M, de Sire A, Colangelo M, Zagaria D, Grassi FA, Rena O, Conte P, Neri P, Carriero A, Sacchetti GM, Penna F, Caretti G, Ferraro E. Sarcopenia Diagnosis: Reliability of the Ultrasound Assessment of the Tibialis Anterior Muscle as an Alternative Evaluation Tool. Diagnostics (Basel). 2021 Nov 21;11(11):2158. doi: 10.3390/diagnostics11112158."

AUTHORS’ RESPONSE: Thank you for this suggested resource. We have added the following information:

“(…) Previous studies have shown good reliability in the ultrasound evaluation of muscles in the lower (2, 3) and upper (4, 5) extremities.” (Introduction, lines 76-77).

Materials and Methods:

REVIEWER COMMENT: Please, specify more in detail the inclusion criteria for the patients and the follow-up in this section. Moreover, it could be interesting to know the rehabilitation program that patient underwent in post-acute care. 

AUTHORS’ RESPONSE:  To provide a better understanding of our work, we have rewritten the subsection regarding inclusion criteria and added a subsection on study procedures to provide more information on the rehabilitation program:

“Inclusion criteria:  1) men and women aged 65 years or older; 2) admitted to a post-acute care unit after hospitalization due to acute illness lasting at least 2 weeks; and 3) cognitive status that permitted understanding study procedures and interventions (Mini-Mental State Examination ≥21/30) (6). Patients with pre-existing conditions that may compromise muscle assessment were excluded (i.e., paresis of the lower limbs due to any neurological disorder, systemic connective tissue disorders, hypo- or hyperthyroidism, lower limb edema at thigh level, systemic atrophies primarily affecting the central nervous system, end-stage renal disease, and end-of-life diseases in palliative care)”. (Materials and Methods, lines 107-114).

“Study procedures; Eligible participants were assessed by a researcher who was in charge of baseline and weekly assessments. Patients in post-acute care underwent an individualized rehabilitation program according to their functional needs (one hour each day, five days a week during hospitalization, usually two weeks). The programs generally focused on improving mobility, starting with the maintenance of joint range, postures, transfers, standing, and walking, followed by progressive muscle strength and endurance exercises.” (Materials and Methods, lines 217-218).

REVIEWER COMMENT: Moreover, a punctual demographics characteristics, such as lab test (total protein, vitamin d status etc.)  and pharmacological treatments is mandatory to better define the population. Moreover, have your population experience of falls or anamnestic fragility fracture? please, specify.

AUTHORS’ RESPONSE: We have included data regarding laboratory values and pharmacological treatment. Additionally, following your suggestion, we have included information on previous fragility fracture and with the data from the SARC-F scale we have been able to add the data on falls in the last year, as follows:

“2.3.9. Laboratory values:  Nutritional values were screened through albumin and prealbumin, and inflammation through C-reactive protein (CRP); vitamin D levels were also evaluated. A blood sample was collected on day of admission to the unit” (Materials and Methods, lines 193-196).

“(…) Twenty participants (50%) had low albumin levels, 24 (60%) had low prealbumin, 30 (75%) had low vitamin D levels, and 37 (93%) showed high levels of CRP. Baseline demographic and clinical characteristics are displayed in Table 2.” (Results, lines 251-254).

“2.3.10. Pharmacological treatment: Non-steroidal anti-inflammatory drugs and drugs relevant to muscle mass, such as glucocorticoids, allopurinol, statins, insulin, and angiotensin II receptor blockers (7-9), were recorded at admission to post-acute care (Materials and Methods, lines 198-201).

“(…) Baseline demographic and clinical characteristics are displayed in Table 2.” (Results, Table 2, pages 7-8).

“(…) Falls in the last year were also recorded, as well as previous fragility fracture, defined as a pathological fracture resulting either from minimal trauma or no identifiable trauma at all (10, 11) (…)”. (Materials and Methods, lines 208-210).

“(…) Baseline demographic and clinical characteristics are displayed in Table 2.” (Results, Table 2, pages 7-8).

REVIEWER COMMENT: The statistical analysis can be improved with analysis more specific to corroborate the association in outcome, such as a multivariate analysis corrected with age, comorbidity and sex. It would be desirable also correction with lab test and pharmacological treatment, if available. 

AUTHORS’ RESPONSE: Thank you very much for your insight. Given the sample size, we used the Student t-test for repeated measures instead of a multivariate analysis. Nevertheless, since the bivariate analyses performed only provided significant results for men, multivariate analysis in our study could not reliably explain a given relationship (even with those significant results). Therefore, we consulted an expert in biostatistics at our biomedical research institute who suggested using one-way ANOVA to compare the two follow-up weeks and multiple comparison post-test using the Dunnett test. (results included in Table 3):

“(…) Changes in outcomes variables, adjusted by baseline values, were assessed by one-way analysis of variance (ANOVA) and multiple comparison post-test with Dunnett's test. (…).” (Materials and Methods, lines 230-232).

“(…) Table 3 shows changes in MT and CSA of the rectus femoris, adjusted by their baseline values. In the first week’s follow-up, men had a significant increase in both MT (0.9 mm [95%CI 0.3 to 1.4], p=0.003] and CSA (0.4 cm2 [95%CI 0.1 to 0.6], p=0.007). At the end of the second week, men again showed a significant increase in MT (0.7 mm [95%CI 0.1 to 1.4], p=0.036) and CSA (0.6 mm [95%CI 0.01 to 1.2], p=0.048). Although an upwards trend in MT and CSA was observed in women throughout the follow-up, these changes were not statistically significant.” (Results, lines 270-276).

Results:

REVIEWER COMMENT: Reading the table 3, It seems that HG, free fat mass and gait speed are inversely associated to cross sectional area at 3 weeks. Please, specify this point in discussion

AUTHORS’ RESPONSE: We did not posit any association between HG, fat-free mass and gait speed because statistical significance was not achieved. In the revised manuscript, we have removed data regarding the 3-week follow-up as most of the patients were discharged home after 2 weeks. We consulted our research insitute’s biostatistician, who recommended against including data for such a small sample.

Discussion:

REVIEWER COMMENT: The discussion is lack of information. I suggest to improve it with a robust literature to support your results and, if applicable, compare them with other similar study (e.g. US in sarcopenia).

AUTHORS’ RESPONSE: Thank you for your suggestion. We have added a reference that suggests MT cut-off points for sarcopenia diagnosis. Although there is no consensus on the use of these cut-off points, our patients had MT values below these levels upon admission to the post-acute care unit. We have also included a paragraph regarding the observed sex-related differences, putting forward plausible explanations for this finding.

“The MT value, followed by CSA, are considered the simplest and most reproducible ultrasound parameters for muscle mass assessment (12). Values for rectus femoris MT of 16 mm in women and 20 mm in men have been suggested as cut-off points in the diagnosis of sarcopenia (13). Although there is no definite consensus on the use of these values, upon admission to post-acute care our patients presented with a mean MT of 13.2 (SD 2.9) mm in women and 14.9 (SD 3.4) mm in men. By the proposed definition, all would qualify for a sarcopenia diagnosis. Even though the loss of muscle mass during the acute disease could not be quantified in our participants, a significant increase in MT and CSA of the rectus femoris was observed in men after the first week in the post-acute unit, with further changes after the second week. Women showed an upwards trend in MT and CSA throughout the follow-up, but these changes did not reach statistical significance.” (Discussion, lines 325-335).

“In our study, there no were baseline differences in age, comorbidity, or frailty status between men and women. A study comparing changes in muscle strength and MT in response to exercise in young adults suggests that MT response in women may have a different time course, with a delay in muscle strength recovery, compared to men (14). However, no clear explanation has been established for these sex-related differences and other factors also may have contributed, such as sex hormones, inflammatory response, sedentary lifestyle, and dietary habits, among others (15).” (Discussion, lines 336-342).

“(…) These findings are of important clinical relevance for patients at risk of frailty, which have proven to benefit from interventions that help to improve their muscle mass (16). Therefore, our study highlights the need for adequate identification of frailty and sarcopenia in this population and supports the importance of including muscle mass monitoring within the comprehensive geriatric assessment in hospitalized older adults (17) to target interventions (e.g. exercise) that enable to attenuate or prevent muscle wasting (18).” (Discussion, lines 347-353).

ADDED REFERENCES

  1. de Sire A, Ferrillo M, Lippi L, Agostini F, de Sire R, Ferrara PE, et al. Sarcopenic Dysphagia, Malnutrition, and Oral Frailty in Elderly: A Comprehensive Review. Nutrients. 2022;14(5).
  2. Leigheb M, de Sire A, Colangelo M, Zagaria D, Grassi FA, Rena O, et al. Sarcopenia Diagnosis: Reliability of the Ultrasound Assessment of the Tibialis Anterior Muscle as an Alternative Evaluation Tool. Diagnostics (Basel). 2021;11(11).
  3. Hammond K, Mampilly J, Laghi FA, Goyal A, Collins EG, McBurney C, et al. Validity and reliability of rectus femoris ultrasound measurements: Comparison of curved-array and linear-array transducers. J Rehabil Res Dev. 2014;51(7):1155-64.
  4. Meza-Valderrama D, Sánchez-Rodríguez D, Perkisas S, Duran X, Bastijns S, Dávalos-Yerovi V, et al. The feasibility and reliability of measuring forearm muscle thickness by ultrasound in a geriatric inpatient setting: a cross-sectional pilot study. BMC Geriatr. 2022;22(1):137.
  5. Pirri C, Pirri N, Porzionato A, Boscolo-Berto R, De Caro R, Stecco C. Inter- and Intra-Rater Reliability of Ultrasound Measurements of Superficial and Deep Fasciae Thickness in Upper Limb. Diagnostics (Basel). 2022;12(9).
  6. Crum RM, Anthony JC, Bassett SS, Folstein MF. Population-based norms for the Mini-Mental State Examination by age and educational level. Jama. 1993;269(18):2386-91.
  7. Martín AI, Priego T, López-Calderón A. Hormones and Muscle Atrophy. In: Xiao J, editor. Muscle Atrophy. Singapore: Springer Singapore; 2018. p. 207-33.
  8. Ferrando B, Olaso-Gonzalez G, Sebastia V, Viosca E, Gomez-Cabrera MC, Viña J. Alopurinol y su papel en el tratamiento de la sarcopenia. Revista Española de Geriatría y Gerontología. 2014;49(6):292-8.
  9. Sartiani L, Spinelli V, Laurino A, Blescia S, Raimondi L, Cerbai E, et al. Pharmacological perspectives in sarcopenia: a potential role for renin-angiotensin system blockers? Clin Cases Miner Bone Metab. 2015;12(2):135-8.
  10. Brown JP, Josse RG. 2002 clinical practice guidelines for the diagnosis and management of osteoporosis in Canada. Cmaj. 2002;167(10 Suppl):S1-34.
  11. Borgström F, Karlsson L, Ortsäter G, Norton N, Halbout P, Cooper C, et al. Fragility fractures in Europe: burden, management and opportunities. Arch Osteoporos. 2020;15(1):59.
  12. Ticinesi A, Meschi T, Narici MV, Lauretani F, Maggio M. Muscle Ultrasound and Sarcopenia in Older Individuals: A Clinical Perspective. J Am Med Dir Assoc. 2017;18(4):290-300.
  13. Minetto MA, Caresio C, Menapace T, Hajdarevic A, Marchini A, Molinari F, et al. Ultrasound-Based Detection of Low Muscle Mass for Diagnosis of Sarcopenia in Older Adults. Pm r. 2016;8(5):453-62.

  1. Fujiwara K, Asai H, Toyama H, Kunita K, Yaguchi C, Kiyota N, et al. Changes in muscle thickness of gastrocnemius and soleus associated with age and sex. Aging Clin Exp Res. 2010;22(1):24-30.
  2. Liao CD, Chen HC, Huang SW, Liou TH. The Role of Muscle Mass Gain Following Protein Supplementation Plus Exercise Therapy in Older Adults with Sarcopenia and Frailty Risks: A Systematic Review and Meta-Regression Analysis of Randomized Trials. Nutrients. 2019;11(8).

Reviewer 4 Report

Manuscript ID: ijerph-1997320-peer-review-v1

Manuscript title: Musculoskeletal Ultrasound Shows Muscle Mass Changes during Post-acute Care Hospitalization in Older Adults: A Prospective Cohort Study

Comments

This manuscript reports a study designed to use musculoskeletal ultrasound to prospectively assess changes in muscle thickness (MT) and cross-sectional area (CSA) of the rectus femoris muscle in older patients admitted in a post-acute care unit during a 3-week follow-up. Secondarily, the authors also compared MT and CSA in patients with frailty and pre-frailty and analyzed the correlations of MT and CSA of the rectus femoris muscle with current sarcopenia diagnostic criteria.

The Introduction section seems well-written, with a concise rationale for the study. Previous studies are briefly mentioned. The study aims are clear and adequate for the chosen study design (observational longitudinal). The manuscript is reported following adequate guidelines (STROBE). The Methods seem well described in sufficient details, although some additional inf regarding US measurement may be interesting. Statistical analysis may be improved and the results may need revision.

Major comment

1. 1. Introduction. I missed a strong rationale why the RF muscle is of clinically or methodologically relevant.

2. 2.5 Statistics. Related sample t-tests and analyses of variance could be replaced by other tests for adjustment for covariates (e.g., age and sex) since groups are not randomized due to the study design.

Minor comments

1. 2.2 Eligibility criteria, lines 93-94. Please report the period for admission of consecutive patients for eligibility assessment.

2. 2.3.1 Muscle thickness and cross-sectional area, lines 102-117. Please describe in brief the procedures for measurement of CSA as cited in [19,20]. In particular, how the rectus femoris contour is obtained/draw for area calculation. 

3. 2.3.9. Demographic and clinical characteristics, lines 184-189. Other clinical characteristics may be relevant to characterize the sample. For instance, whether participants were bed restrained; received invasive ventilation and/or other rehabilitation interventions. Consider adding some of these info. Nonetheless, I acknowledge such information may not be available at this point but should be discussed.

Author Response

AUTHORS RESPONSE TO REVIEWERS

We sincerely appreciate all your valuable comments and suggestions, which helped us to improve the quality of the manuscript. We attach a point-by-point response to those comments and the revised manuscript with all changes highlighted.

Reviewer comments:

This manuscript reports a study designed to use musculoskeletal ultrasound to prospectively assess changes in muscle thickness (MT) and cross-sectional area (CSA) of the rectus femoris muscle in older patients admitted in a post-acute care unit during a 3-week follow-up. Secondarily, the authors also compared MT and CSA in patients with frailty and pre-frailty and analyzed the correlations of MT and CSA of the rectus femoris muscle with current sarcopenia diagnostic criteria.

 The Introduction section seems well-written, with a concise rationale for the study. Previous studies are briefly mentioned. The study aims are clear and adequate for the chosen study design (observational longitudinal). The manuscript is reported following adequate guidelines (STROBE). The Methods seem well described in sufficient details, although some additional inf regarding US measurement may be interesting. Statistical analysis may be improved and the results may need revision.

Major comments

Introduction

REVIEWER COMMENT: I missed a strong rationale why the RF muscle is of clinically or methodologically relevant.

AUTHORS’ RESPONSEWe have further developed the rationale for the assessment of the rectus femoris muscle as follows:

“Previous studies have shown good reliability in the ultrasound evaluation of muscles in the lower (1, 2) and upper (3, 4) extremities. Ultrasound measurements of rectus femoris, as cross-sectional area (CSA) or muscle thickness (MT), are reliable, simple and robust tools to detect low muscle mass (5, 6) and accurately predict sarcopenia in older patients (7, 8) with high sensitivity and specificity (9). CSA and MT are linearly related to maximal voluntary contraction strength in patients with chronic respiratory issues, and are also associated with a high to very high prognostic risk in diabetic kidney disease (10). These measurements assess muscle wasting in hemodialysis patients with protein energy wasting (11) and in patients with spinal cord injury (12). They also predict adverse outcomes in patients discharged from a surgical intensive care unit (13), and can discriminate between sarcopenia and non-sarcopenia states (13, 14) (…). (Introduction, lines 76-86).

Statistics

REVIEWER COMMENT: Related sample t-tests and analyses of variance could be replaced by other tests for adjustment for covariates (e.g., age and sex) since groups are not randomized due to the study design.

AUTHORS RESPONSE: Thank you very much for your insight. Given the small sample size, we used the Student t-test for repeated measures instead of a multivariate analysis. Since the bivariate analyses performed only provided significant results for men, multivariate analysis in our study could not reliably explain a given relationship (even with those significant results). We have consulted an expert in biostatistics at our biomedical research institute who suggested using one-way ANOVA to compare the two follow-up weeks and multiple comparison post-test using Dunnett’s test. (results included in Table 3):

“(…) Changes in outcomes variables, adjusted by baseline values, were assessed by one-way analysis of variance (ANOVA) and multiple comparison post-test with Dunnett's test. (…).” (Materials and Methods, lines 230-232).

“(…) Table 3 shows changes in MT and CSA of the rectus femoris, adjusted by their baseline values. In the first week’s follow-up, men had a significant increase in both MT (0.9 mm [95%CI 0.3 to 1.4], p=0.003] and CSA (0.4 cm2 [95%CI 0.1 to 0.6], p=0.007). At the end of the second week, men again showed a significant increase in MT (0.7 mm [95%CI 0.1 to 1.4], p=0.036) and CSA (0.6 mm [95%CI 0.01 to 1.2], p=0.048). Although an upwards trend in MT and CSA was observed in women throughout the follow-up, these changes were not statistically significant.” (Results, lines 269-275).

Minor comments 

REVIEWER COMMENT: Eligibility criteria, lines 93-94. Please report the period for admission of consecutive patients for eligibility assessment.

AUTHORS’ RESPONSE: Thank you. We have added this information as follows:

“(…) The study was conducted in the post-acute care unit of a university hospital in Barcelona (Catalonia, Spain) from October 2019 to March 2021.” (Materials and Methods, lines 103-105).

REVIEWER COMMENT: Muscle thickness and cross-sectional area, lines 102-117. Please describe in brief the procedures for measurement of CSA as cited in [19,20]. How the rectus femoris contour is obtained/draw for area calculation. 

AUTHORS’ RESPONSE: We have developed a new table describing the measurement protocol (Table 1, Materials and Methods, page 4).

REVIEWER COMMENT: Demographic and clinical characteristics, lines 184-189. Other clinical characteristics may be relevant to characterize the sample. For instance, whether participants were bed restrained; received invasive ventilation and/or other rehabilitation interventions. Consider adding some of this info. Nonetheless, I acknowledge such information may not be available at this point but should be discussed.

AUTHORS’ RESPONSE: We have included data regarding laboratory tests and pharmacological treatment. The reviewer is correct that we do not have access to complete data on other aspects of patient hospitalization. However, we have included information on previous fragility fracture, and with the data from the SARC-F scale we have been able to add the data on falls in the last year, as follows:

“2.3.9. Laboratory values:  Nutritional values were screened through albumin and prealbumin, and inflammation through C-reactive protein (CRP); vitamin D levels were also evaluated. A blood sample was collected on day of admission to the unit” (Materials and Methods, lines 193-196).

“(…) Twenty participants (50%) had low albumin levels, 24 (60%) had low prealbumin, 30 (75%) had low vitamin D levels, and 37 (93%) showed high levels of CRP. Baseline demographic and clinical characteristics are displayed in Table 2.” (Results, lines 251-254).

“2.3.10. Pharmacological treatment: Non-steroidal anti-inflammatory drugs and drugs relevant to muscle mass, such as glucocorticoids, allopurinol, statins, insulin, and angiotensin II receptor blockers (15-17), were recorded at admission to post-acute care (Materials and Methods, lines 198-201).

“(…) Baseline demographic and clinical characteristics are displayed in Table 2.” (Results, Table 2, pages 7-8).

“(…) Falls in the last year were also recorded, as well as previous fragility fracture, defined as a pathological fracture resulting either from minimal trauma or no identifiable trauma at all (18, 19) (…)”. (Materials and Methods, lines 208-210).

“(…) Baseline demographic and clinical characteristics are displayed in Table 2.” (Results, Table 2, pages 7-8).

ADDED REFERENCES

  1. Leigheb M, de Sire A, Colangelo M, Zagaria D, Grassi FA, Rena O, et al. Sarcopenia Diagnosis: Reliability of the Ultrasound Assessment of the Tibialis Anterior Muscle as an Alternative Evaluation Tool. Diagnostics (Basel). 2021;11(11).
  2. Hammond K, Mampilly J, Laghi FA, Goyal A, Collins EG, McBurney C, et al. Validity and reliability of rectus femoris ultrasound measurements: Comparison of curved-array and linear-array transducers. J Rehabil Res Dev. 2014;51(7):1155-64.
  3. Meza-Valderrama D, Sánchez-Rodríguez D, Perkisas S, Duran X, Bastijns S, Dávalos-Yerovi V, et al. The feasibility and reliability of measuring forearm muscle thickness by ultrasound in a geriatric inpatient setting: a cross-sectional pilot study. BMC Geriatr. 2022;22(1):137.
  4. Pirri C, Pirri N, Porzionato A, Boscolo-Berto R, De Caro R, Stecco C. Inter- and Intra-Rater Reliability of Ultrasound Measurements of Superficial and Deep Fasciae Thickness in Upper Limb. Diagnostics (Basel). 2022;12(9).
  5. Fukumoto Y, Ikezoe T, Taniguchi M, Yamada Y, Sawano S, Minani S, et al. Cut-off Values for Lower Limb Muscle Thickness to Detect Low Muscle Mass for Sarcopenia in Older Adults. Clin Interv Aging. 2021;16:1215-22.
  6. Berger J, Bunout D, Barrera G, de la Maza MP, Henriquez S, Leiva L, et al. Rectus femoris (RF) ultrasound for the assessment of muscle mass in older people. Arch Gerontol Geriatr. 2015;61(1):33-8.

  1. Lin X, Chen Z, Huang H, Zhong J, Xu L. Diabetic kidney disease progression is associated with decreased lower-limb muscle mass and increased visceral fat area in T2DM patients. Front Endocrinol (Lausanne). 2022;13:1002118.
  2. Sahathevan S, Khor BH, Singh BKS, Sabatino A, Fiaccadori E, Daud ZAM, et al. Association of Ultrasound-Derived Metrics of the Quadriceps Muscle with Protein Energy Wasting in Hemodialysis Patients: A Multicenter Cross-Sectional Study. Nutrients. 2020;12(11).
  3. Tay MRJ, Kong KH. Ultrasound Measurements of Rectus Femoris and Locomotor Outcomes in Patients with Spinal Cord Injury. Life (Basel). 2022;12(7).
  4. Mueller N, Murthy S, Tainter CR, Lee J, Riddell K, Fintelmann FJ, et al. Can Sarcopenia Quantified by Ultrasound of the Rectus Femoris Muscle Predict Adverse Outcome of Surgical Intensive Care Unit Patients as well as Frailty? A Prospective, Observational Cohort Study. Ann Surg. 2016;264(6):1116-24.
  5. Rustani K, Kundisova L, Capecchi PL, Nante N, Bicchi M. Ultrasound measurement of rectus femoris muscle thickness as a quick screening test for sarcopenia assessment. Arch Gerontol Geriatr. 2019;83:151-4.
  6. Martín AI, Priego T, López-Calderón A. Hormones and Muscle Atrophy. In: Xiao J, editor. Muscle Atrophy. Singapore: Springer Singapore; 2018. p. 207-33.
  7. Ferrando B, Olaso-Gonzalez G, Sebastia V, Viosca E, Gomez-Cabrera MC, Viña J. Alopurinol y su papel en el tratamiento de la sarcopenia. Revista Española de Geriatría y Gerontología. 2014;49(6):292-8.
  8. Sartiani L, Spinelli V, Laurino A, Blescia S, Raimondi L, Cerbai E, et al. Pharmacological perspectives in sarcopenia: a potential role for renin-angiotensin system blockers? Clin Cases Miner Bone Metab. 2015;12(2):135-8.
  9. Brown JP, Josse RG. 2002 clinical practice guidelines for the diagnosis and management of osteoporosis in Canada. Cmaj. 2002;167(10 Suppl):S1-34.
  10. Borgström F, Karlsson L, Ortsäter G, Norton N, Halbout P, Cooper C, et al. Fragility fractures in Europe: burden, management and opportunities. Arch Osteoporos. 2020;15(1):59.

Round 2

Reviewer 1 Report

All considerations have been addressed by the authors, and the manuscript has been considerably improved. Congratulations.

Reviewer 3 Report

At the light of my knowledge, the paper is suitable for fully publication in journal